# In Situ Tensile Observation of the Effect of Annealing on Fracture Behavior of Laser Additive Manufactured Titanium Alloy

**DOI:** 10.3390/ma16113973

**Published:** 2023-05-25

**Authors:** Ming Xu, Huaixue Li, Qi Liu, Feihu Shan, Yongxin Zhang, Wei Guo, Fei Li, Hongqiang Zhang

**Affiliations:** 1AVIC Manufacturing Technology Institute, Beijing 100024, China; 2Key Laboratory of Aeronautical Technology on Additive Manufacturing, Beijing 100024, China; 3School of Mechanical Engineering and Automation, Beihang University, Beijing 100191, China

**Keywords:** additive manufacturing, titanium alloys, annealing, in situ tensile, fracture mechanism

## Abstract

The use safety of laser additive manufactured (LAM) titanium alloys is closely related to each one’s fracture failure mode. In this study, in situ tensile tests were carried out to study the deformation and fracture mechanisms of LAM Ti6Al4V titanium alloy before and after annealing treatment. The results showed that plastic deformation promoted slip bands to occur inside the α phase and shear bands to generate along the α/β interface. In the as-built sample, cracks initiated in the equiaxed grains and propagated along the columnar grain boundary, showing a mixed fracture mode. However, it transformed into a transgranular fracture after annealing treatment. The widmanstatten α phase acted as a barrier for slip movement and improved the crack resistance of grain boundaries.

## 1. Introduction

In recent years, laser additive manufacturing (LAM) technology has set off a manufacturing revolution in the aerospace field [1,2]. This technology uses a high-energy laser beam to melt the alloy powder. The molten pool is continuously formed and then rapidly solidified, thus depositing layer by layer to near-net-forming metal components [3]. Titanium alloy, as an important structural metal, has advantages of high strength, high toughness, low density and good corrosion resistance [4,5,6]. Using LAM to prepare titanium alloy parts is expected to obtain key components with high performance and high quality.

Titanium alloy parts experienced a high temperature gradient and a high cooling rate during the LAM process, resulting in a large difference in the microstructure from traditional materials. Usually, there are α phase, martensite α’ phase or a mixture of both in the prior β grains, and the continuous α phase was also embedded along the prior β grain boundary [7,8,9]. Carroll et al. [10] reported that the grain boundary α phase and prior β grain morphologies caused the anisotropic mechanical properties of additively manufactured titanium alloys. In addition, the α phases with high strength and low toughness lead to the mismatch of strength and toughness of the formed parts [11]. By the heat treatment process, the morphology, size and proportion of the phases can be effectively controlled, thereby obtaining good mechanical properties [12,13,14,15]. Yadroitsev et al. [16] reported that a large number of spherical α phases were produced near the β phase transition temperature. Zhao et al. [17] obtained two types of microstructures of basket-weave and colony structures by controlling the cooling rate. The tensile results showed that the former had higher strength and toughness, which may be attributed to the lamellar α phase in the basket-weave structure effectively reducing the dislocation slip length and dispersing local stress concentration. However, due to the lack of observation of the microstructure evolution during the tensile process, the deformation and failure mechanism are still unclear. In situ tensile test can directly observe the deformation process of materials by controlling the loading displacement, which provides an effective method for an on-line observation of the dynamic damage of the microstructure [18,19].

In this work, in situ tensile tests were carried out to observe the dynamic tensile process of LAM Ti6Al4V. This study aims to reveal the damage and fracture behavior of LAM Ti6Al4V and explore the mechanism of performance improvement after annealing treatment. The results are of great significance for the performance strengthening design and the service safety evaluation of LAM Ti6Al4V.

## 2. Materials and Methods

Ti6Al4V samples were prepared by a laser-directed energy deposition (LDED) system (LDD-6000, AVIC Manufacturing Technology Institute, Beijing, China) with the diameter of powder particles ranging from 80 μm to 120 μm. The chemical composition (wt. %) was measured as Al: 6.75, V: 4.5, O: 0.20, H: 0.015, N: 0.05, C: 0.10, Fe: 0.30 and Ti balance. During the LDED process, the laser power was 3 kW, the scanning speed was 900 mm/min and the powder feed rate was 50 g/min. In order to reduce the anisotropy, the laser scanning path with a 45° angle was adopted to build the parts layer by layer (Figure 1a). Subsequently, an annealing treatment at 550 °C for 2 h was performed in a vacuum brazing furnace (VBF-113, Shenzhen Vacuum Technology Co., Ltd., Shenyang, China). The detailed parameters of the annealing treatment strategy are listed in Table 1.

The metallographic samples were machined via wire-electrode cutting. After mechanical polishing and etching in Kroll reagent (1 mL HF + 3 mL HNO_3_ + 7 mL H_2_O), the microstructures were characterized using an optical microscope (OM, Scope.A1, Carl Zeiss, Oberkochen, Germany) and a scanning electron microscope (SEM, JSM-7001F, JEOL Ltd., Tokyo, Japan). The in situ tensile samples were cut transversely on the XZ plane. The size of tensile samples is shown in Figure 1b. The tensile samples were also polished and etched to observe the microstructure evolution under SEM (Joel IT300lv, JEOL Ltd., Tokyo, Japan). During the in situ test, the tensile displacement was suspended several times for SEM imaging. The tensile rate was kept at 0.1 mm/min.

## 3. Results and Discussions

### 3.1. Microstructures

Figure 2 shows the typical microstructures of as-built and annealed Ti6Al4V samples. In the as-built sample, the prior β columnar crystal was epitaxially grown along the build direction, and the equiaxed β grains were distributed at the top of the deposited layer (Figure 2a). This difference in grain morphology should be related to the solidification process of the molten pool [20]. At the bottom of the molten pool, a large temperature gradient was caused by rapid heat loss along the previous deposited layer, which promoted the preferential growth of the β phase along the <100> direction to form columnar grains [21,22]. However, on the surface of the molten pool, adsorbed or incompletely melted powder particles induced heterogeneous nucleation, resulting in the formation of equiaxed grains [23]. After annealing treatment in Figure 2b, the macroscopic characteristics did not change significantly due to the heating temperature’s being lower than the β transition temperature (980 °C).

Various types of α phases were formed during the LAM solidification process as shown in Figure 3. The grain boundary α phases (α_GB_) were continuously embedded along the prior β grains (Figure 3a). The widmanstatten α phases (α_W_) nucleated at α_GB_ and grew into the β grains (Figure 3b). Because they inherited the orientation characteristics of the parent phase α_GB_, the α_W_ phase exhibited a parallel-arranged plate structure [17,24]. Inside prior β grains, the nucleation and growth of α phase were promoted by supercooling until forming an α colony region where α plates shared the same orientation (Figure 3c). After annealing treatment (Figure 3d–f), α_GB_ and α_W_ were not changed significantly, while the size of the α colony decreased slightly due to the fracture of the long α plate [25].

### 3.2. In Situ Tensile Properties

Figure 4 shows the stress–strain curves measured with in situ tensile tests. The gaps on the curve represent the in situ observation process after a suspension of tensile displacement. The ultimate tensile strength of the as-built sample was 899.7 MPa, and the elongation was 13.9%. After annealing treatment, the ultimate tensile strength increased to 966.7 MPa and the elongation increased to 14.8%. This improvement in tensile properties was closely related to the fracture form, which was further studied in the following work.

### 3.3. Fracture Process

#### 3.3.1. Mixed Fracture in as-Built Sample

Figure 5 shows the macroscopic morphology of the as-built sample during the in situ tensile process. The tensile strain of 0%, 4%, 6%, 8%, 11%, 12%, 13% and the fracture moment correspond to the Figure 5a–g and Figure 4A, respectively. Before 4% strain, the sample was in the elastic deformation stage; thus, the macroscopic surface and microstructure did not change significantly (Figure 5a,b). When the tensile strain reached 6 %, the sample began to undergo plastic deformation, and the slip line was generated (Figure 5c). As the tensile displacement continue, the significant increase in the number of slip lines indicated an intensified plastic deformation (Figure 5d). The magnification diagram showed that the plastic strain made the α plates convex and concave, resulting in the fluctuation of the sample surface. Subsequently, shear bands were observed within the prior β grains (see Figure 5e,f), which usually occurred along the macroscopic plane with the maximum shear stress, i.e., 45° to the tensile direction [26]. Due to the obvious depression caused by the deformation of the α phase in the crystal, the profile of the β grain boundary became protruding (see the inset in Figure 5e). Especially at the deposited line, high-density grain boundaries were clearer around equiaxed grains (see the inset in Figure 5g). Further tensile loading, the sample was fractured, showing a mixed fracture mode with cracks straight in the middle and curved on both sides (Figure 5h).

Figure 6 shows the interaction between slip bands and grain boundaries during uniaxial tension in the region near the fracture. At low strain, the grain surface was relatively flat (Figure 6a), while slip bands appeared between the layers of α phases (see the upper right inset). As the strain reached 8%, the resolved shear stress promoted the formation of shear band (Figure 6b). It was worth noting that the direction of shear bands in adjacent grains were different due to the difference in the orientation of prior β grains [27]. As the deformation intensified, the gradual depression in the crystal was accompanied by a significant increase in the number and length of slip bands (Figure 6c,d). At this time, α_GB_ phase beard the open stress to form a step in Figure 6e [28]. As the damage accumulated, a microcrack initiated along the α_GB_ phase, providing a fast channel for crack propagation (Figure 6f).

Although microcracks were easily formed along α_GB_ phases, this was not the initiation point of the main crack in the as-built sample. Figure 7 captures the continuous process of the fracture failure. It can be seen from Figure 7a that the main crack originated from the interior of the equiaxed grains at the deposited line. The existence of high-density grain boundaries limited the slip distance, resulting in a great accumulation of strain in equiaxed grains. In the enlarged Figure 7b, the equiaxed grain were marked as grain I and II, and the columnar grain were marked as grain III. It can be clearly seen that the main crack initiated along the 45° shear band inside grain I. Subsequently, it expanded upward in a zigzag path, passed through the grain boundary into grain III and continued to grow along the 45° shear band. When the crack tip encountered the grain boundary of grain III, it deflected to expand along the α_GB_ phase (Figure 7c). As the loading continues, the crack grew rapidly until it broke under the open load (Figure 7d). It can be concluded from Figure 7e that the as-built sample underwent a mixed fracture mode—that is to say, cracks initiated inside equiaxed grain and grew along the columnar grain boundary.

#### 3.3.2. Transgranular Fracture after Annealing Treatment

Figure 8 shows the macroscopic morphology of the annealed sample at 0%, 6%, 10%, 13%, 13.5% strain and fracture moment, corresponding to the a-f points in Figure 3b, respectively. In the elastic stage, the sample surface showed no significant change, and the grain boundary did not bulge to form steps (Figure 8a,b). When the strain increased to 10%, local deformation with shear band and slip band morphology occurred preferentially at the edge of the sample due to geometric factors (Figure 8c). This induced crack initiation and propagation along the shear band in further tensile strain (Figure 8d). At 13.5% strain, the crack tip branched to form crack I and crack II (Figure 8e). Crack I developed into the main crack, while crack II ended in the slip band. The crack propagation path after the fracture was shown in Figure 8f. It can be seen that the Z-shaped main crack continuously passed through the interior of the columnar grain and equiaxed grain, showing a typical transgranular fracture path. This indicated that the annealing treatment strengthened the crack resistance of the prior β grain boundary and avoided it becoming a fast path for crack propagation. Ren et al. [29] also reported the phenomenon of crack transgranular growth rather than along the prior β grain boundary.

Figure 9 shows the main crack propagation path and the nearby microstructure changes in the annealed sample. It can be seen that the main crack was far away from the prior β grain boundary (Figure 9a), and a large number of shear bands were distributed in parallel near the main crack (Figure 9b). This induced secondary cracks along the shear band. In the enlarged image (Figure 9c), the slip bands were distributed in the α phase, and the shear bands were preferentially formed along the α/β interface, where the α phase was nearly 45° oriented to the tensile direction. Figure 9d summarized the deformation mechanism under the tensile load. For the α phase with a 45° orientation to the tensile axis, the critical shear stress promoted the shear strain to accumulate along the α/β interface to form shear bands, while for the α phase in other directions, the slip system was activated to form a large number of slip bands [30]. When the strain was transferred to the outside of the grain, the grain boundary remained intact without cracking compared with the as-built sample (Figure 9e). The phenomenon that the slip band was perpendicular to the widmanstatten α_w_ phase was further observed near the grain boundary (Figure 9f). Therefore, it was reasonably speculated that the dense widmanstatten α_w_ phase acted as a barrier for the slip band to move to the grain boundary, which alleviated the strain accumulation and enhanced the crack resistance at the grain boundary (Figure 9g).

### 3.4. Fracture Mechanism before and after Annealing Treatment

Figure 10 summarizes the fracture mechanism of LAM Ti6Al4V titanium alloy during uniaxial tensile process. In the early stage of plastic deformation, resolved shear stress promoted the appearance of slip lines, slip bands and shear bands inside prior β grains [31]. As the loading continued, the slip increased in number and extended to the grain boundary, but the grain boundary acted as a barrier to the slip movement [32,33,34]. Therefore, in the equiaxed grain zone, the high-density grain boundaries aggravated the degree of intragranular plastic strain, which eventually led to the preferential initiation of cracks inside equiaxed grains. At the columnar grains, low-density grain boundaries provided space for long-distance slip, which homogenized intragranular plastic deformation. The steps formed at the grain boundary due to strain accumulation and induced microcracks. Finally, the as-built sample exhibited a mixed fracture mode in which cracks initiated in equiaxed grains and grew along columnar grain boundaries (Figure 10a). After annealing treatment, the crack resistance of the grain boundary was enhanced, which may be due to the recovery of the grain boundary with reduced internal stress after annealing treatment [33]. In addition, the parallel widmanstatten α_w_ phase hindered the slip to the grain boundary and reduced the local strain accumulation. Therefore, the prior β grain boundary had an enhanced crack resistance, resulting in a transgranular fracture mode (Figure 10b).

## 4. Conclusions

In situ tensile experiments were carried out to study the deformation mechanism and fracture model of LAM Ti6Al4V alloy before and after annealing treatment. The main conclusions are as follows:In the plastic deformation stage, slip lines and slip bands were generated inside the α phase, and shear bands were generated along the α/β interface.The as-built sample had a mixed fracture mode. Severe plastic deformation occurred in the equiaxed grains zone due to high-density grain boundaries. Cracks initiated inside the equiaxed grains and propagated along the columnar grain boundaries.The annealing treatment at 550 °C for 2 h could enhance the tensile strength and elongation by changing the failure mode to a transgranular fracture. The widmanstatten α phase prevented slip from moving to the grain boundary and improved the crack resistance of the grain boundary.

## Figures and Tables

**Figure 1 materials-16-03973-f001:**
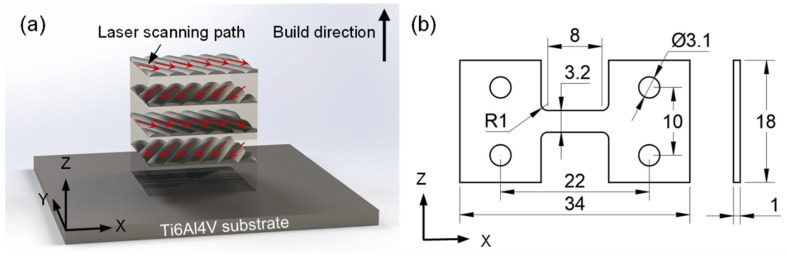
(**a**) LAM process diagram and red arrows representing the laser scanning path at a 45° angle, (**b**) detail dimensions of in situ tensile specimen (unit: mm).

**Figure 2 materials-16-03973-f002:**
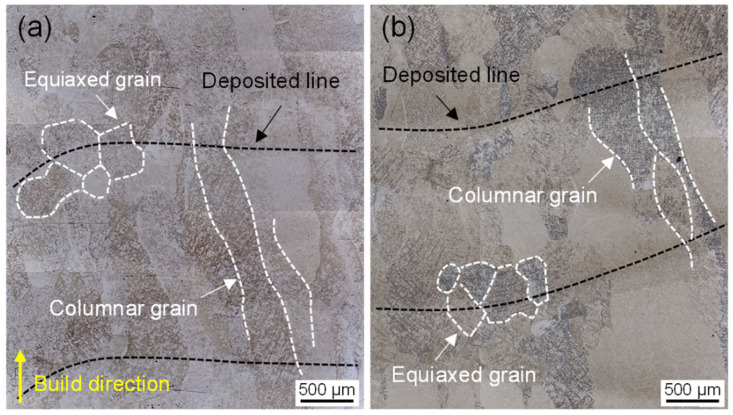
OM micrograph showing the β morphology in LDED Ti6Al4V: (**a**) as-built sample, (**b**) annealed sample.

**Figure 3 materials-16-03973-f003:**
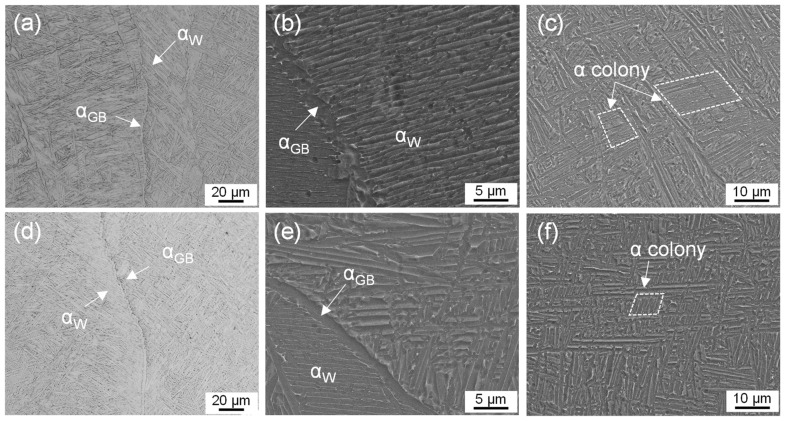
Various types of α phases including α_GB_, α_w_ and α colony: (**a**–**c**) as-built sample, (**d**–**f**) annealed sample.

**Figure 4 materials-16-03973-f004:**
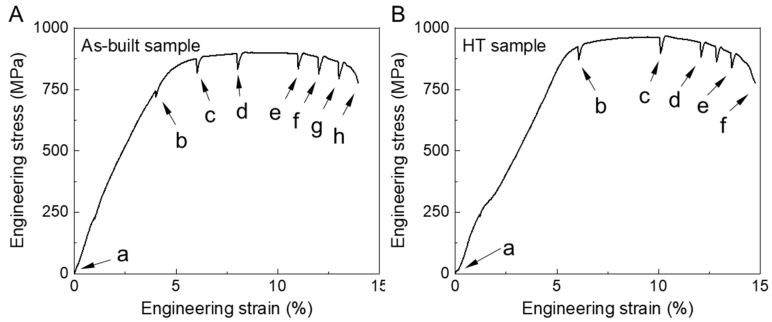
Stress-strain curve for in situ tensile: (**A**) as-built sample, (**B**) annealed sample. Lowercase letters representing the suspension of tensile displacement for microstructure observation.

**Figure 5 materials-16-03973-f005:**
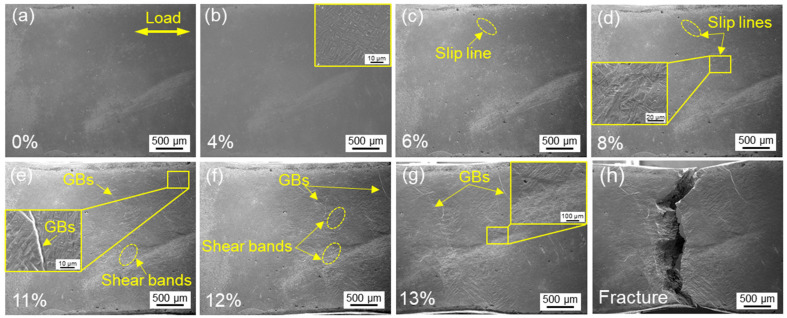
Deformation morphology in as-built samples under different strains: (**a**) 0%, (**b**) 4%, (**c**) 6%, (**d**) 8%, (**e**) 11%, (**f**) 12%, (**g**) 13% and (**h**) fracture moment.

**Figure 6 materials-16-03973-f006:**
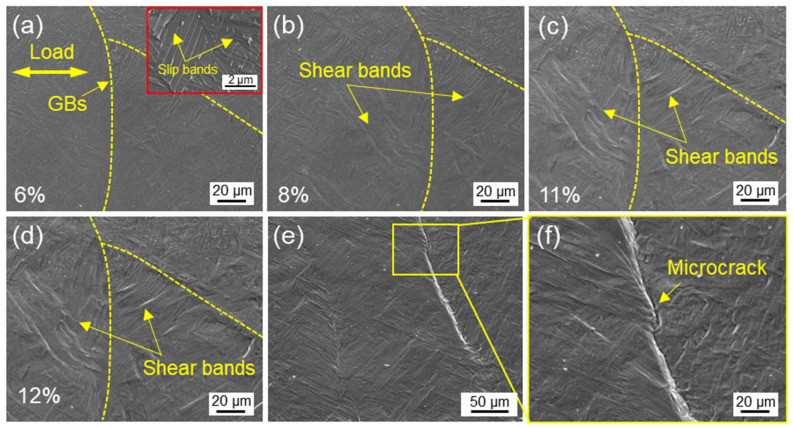
Slip morphology between adjacent grains under (**a**) 6%, (**b**) 8%, (**c**) 11% and (**d**) 12% strain. (**e**) Steps formed along grain boundaries inducing (**f**) crack initiation.

**Figure 7 materials-16-03973-f007:**
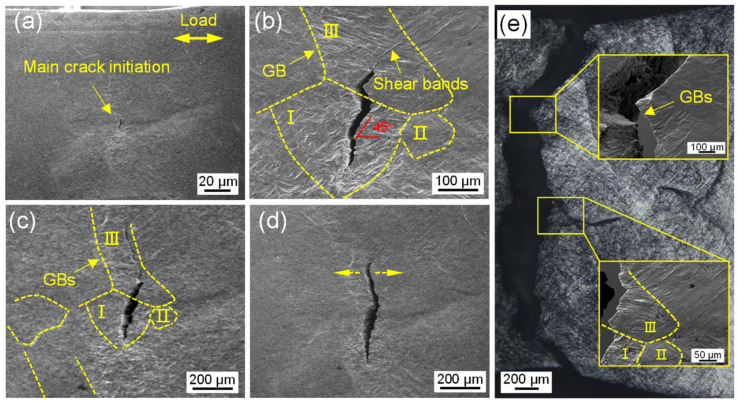
Fracture process for as-built sample: (**a**) crack initiating in equiaxed grains, (**b**) crack passing through the equiaxed-columnar grain boundary, (**c**) crack propagating along the columnar grain boundary, (**d**) crack opening instability and (**e**) fracture morphology.

**Figure 8 materials-16-03973-f008:**
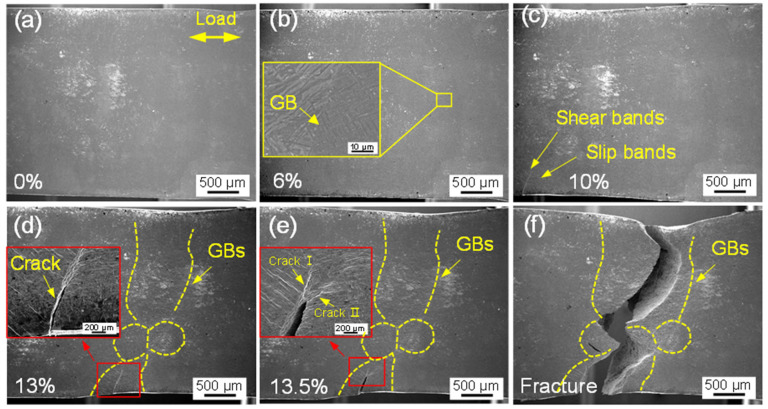
Deformation morphology of annealed sample under different strains: (**a**) 0%, (**b**) 6%, (**c**) 10%, (**d**) 13%, (**e**) 13.5% and (**f**) fracture moment.

**Figure 9 materials-16-03973-f009:**
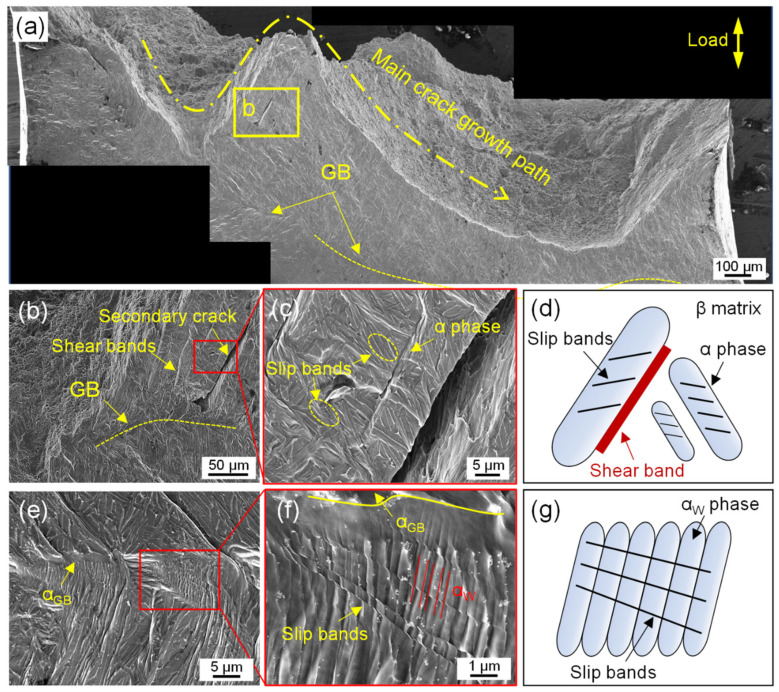
The fracture and nearby microscopic deformation morphology in the annealed sample: (**a**) main crack propagation path, (**b**) secondary crack along the shear band, (**c**) slip and shear band morphology, (**d**) position diagram for slip and shear band, (**e**) complete grain boundary without cracking, (**f**) slip band in widmanstatten α_w_ phase, (**g**) α_w_ phase hindering slip band diagram.

**Figure 10 materials-16-03973-f010:**
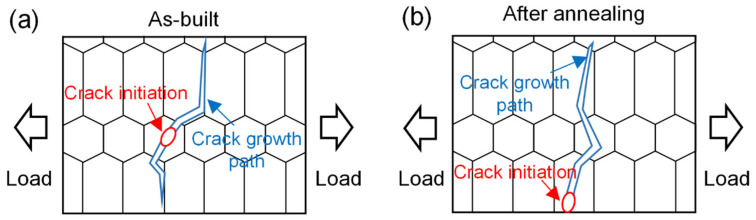
(**a**) Mixed fracture in the as-built sample, (**b**) transgranular fracture in the annealed sample.

**Table 1 materials-16-03973-t001:** Annealing treatment strategy for LAM Ti6Al4V.

Sample	Temperature (°C)	Holding Time (h)	Heating Rate (°C/min)	Cooling Method
As-built	-	-	-	-
Annealed	550	2	10	Furnace cooling

## Data Availability

Data will be made available on request.

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
