# Peer review of "In Situ Tensile Observation of the Effect of Annealing on Fracture Behavior of Laser Additive Manufactured Titanium Alloy"

_materials, 2023, doi:10.3390/ma16113973_

Round 1

Reviewer 1 Report

1.      Take-home message from this work for a LDED practitioner regarding process improvement should be stated clearly in the Conclusion.

2.      Larger and clearer images of Fig. 5(a)-(g) and 8(a)-(f) should be provided.

3.      Provide parameters used for LDED of the tensile specimen.

4.      Provide reference to justify the choice of dimensions of the tensile specimens.

 Correct English. For e.g. change “Titanium alloy parts were experienced high temperature gradient…” to “Titanium alloy parts experienced high temperature gradient…” in page 1; change “This induced that crack initiated and propagated along the shear band…” to “This induced that crack which got initiated and propagated along the shear band…” in page 7 and so on.

Author Response

Reviewer 1#

  1. Take-home message from this work for a LDED practitioner regarding process improvement should be stated clearly in the Conclusion.

√Thanks for your suggestions, we have improved the conclusions as follows: “3. The annealing treatment at 550°C for 2h could enhance the tensile strength and elongation by changing the failure mode to transgranular fracture. The widmanstatten α phase prevented slip from moving to the grain boundary and improved the crack resistance of the grain boundary.”

  1. Larger and clearer images of Fig. 5(a)-(g) and 8(a)-(f) should be provided.

√Thanks to your suggestions. We have added some details and improved the image quality.

Before 4% strain, the sample was in the elastic deformation stage, thus the macro-scopic surface and microstructure did not change significantly (Figs. 5a and b). When the tensile strain reached 6 %, the sample began to undergo plastic deformation, and the slip line was generated (Fig. 5c). As the tensile displacement continue, the significant increase in the number of slip lines indicated an intensified plastic deformation (Fig. 5d). The magnification diagram showed that the plastic strain made the α plates convex and concave, resulting in the fluctuation of the sample surface. Subsequently, shear bands were observed within the prior β grains (see Figs. 5e and f), which usually occurred along the macroscopic plane with the maximum shear stress, i.e., 45° to the tensile direction. Due to the obvious depression caused by the deformation of the α phase in the crystal, the profile of the β grain boundary became protruding (see the in-set in Fig. 5e). Especially at the deposited line, high-density grain boundaries were clearer around equiaxed grains (see the inset in Fig. 5g). Further tensile loading, the sample was fractured, showing a mixed fracture mode with cracks straight in the mid-dle and curved on both sides (Fig. 5h).

Figure 5. Deformation morphology in as-built samples under different strains: (a) 0%, (b) 4%, (c) 6%, (d) 8%, (e) 11%, (f) 12%, (g) 13% and (h) fracture moment.

In the elastic stage, the sample surface showed no significant change, and the grain boundary did not bulge to form steps (Figs. 8a and b). When the strain increased to 10%, local deformation with shear band and slip band morphology occurred preferen-tially at the edge of the sample due to geometric factors (Fig. 8c). This induced crack initiation and propagation along the shear band in further tensile strain (Fig. 8d). At 13.5 % strain, the crack tip branched to form crack I and crack II (Fig. 8e). Crack I de-veloped into the main crack, while crack II ended in the slip band. The crack propaga-tion path after fracture was shown in Fig. 8f. It could be seen that the Z-shaped main crack continuously passed through the interior of the columnar grain and equiaxed grain, showing a typical transgranular fracture path.

Figure 8. Deformation morphology of annealed sample under different strains: (a) 0%, (b) 6%, (c) 10%, (d) 13%, (e) 13.5% and (f) fracture moment.

  1. Provide parameters used for LDED of the tensile specimen.

√Thanks for your suggestions. We have added LDED details: “During the LDED process, the laser power was 3 kW, the scanning speed was 900 mm/min, and the powder feed rate was 50 g/min. In order to reduce the anisotropy, the laser scanning path with a 45° angle was adopt-ed to build the parts layer by layer (Fig. 1a).”

  1. Provide reference to justify the choice of dimensions of the tensile specimens.

√Thanks to your suggestions. The specimen size is provided by Central Iron & Steel Research Institute, and the relevant research results have been published [1].

[1] Xue J, Guo W, Yang J, et al. In-situ observation of microcrack initiation and damage nucleation modes on the HAZ of laser-welded DP1180 joint[J]. Journal of Materials Science & Technology, 2023.

  1. Comments on the Quality of English Language. Correct English. For e.g. change “Titanium alloy parts were experienced high temperature gradient…” to “Titanium alloy parts experienced high temperature gradient…” in page 1; change “This induced that crack initiated and propagated along the shear band…” to “This induced that crack which got initiated and propagated along the shear band…” in page 7 and so on.

√Thanks for your suggestions. We have rewritten the sentences to improve the quality of English language.

Reviewer 2 Report

The article entitled "In-situ tensile observation of the effect of annealing on fracture behavior of laser additive manufactured titanium alloy" deals with the very current topic of the behavior of 3D printed materials. The article is written according to the scientific form in a high-quality, readable manner, and is divided into logical contexts. The research results can be used for other scientific works.

I would have a few comments about the authors and minor flaws in the text.

the authors do not state the printing parameters under which the samples were produced. In doing so, the printing parameters have a major influence and should be listed.

For other authors, they do not indicate the number of samples on which the experiments were performed. And if he gives the results in the text of Figure 4, then the results are given without statistical values.

Figure 3a and 3d have very poor image quality.

There are minor typos in the text, such as:

  the name of the Joel IT 300 lv microscope is not in brackets like the rest of the instruments.

Page 6 is at Fig. 7 is written in small letters.

I recommend the article to be published after performing a minor revision.

Author Response

Reviewer 2#

The article entitled "In-situ tensile observation of the effect of annealing on fracture behavior of laser additive manufactured titanium alloy" deals with the very current topic of the behavior of 3D printed materials. The article is written according to the scientific form in a high-quality, readable manner, and is divided into logical contexts. The research results can be used for other scientific works. I would have a few comments about the authors and minor flaws in the text.

  1. the authors do not state the printing parameters under which the samples were produced. In doing so, the printing parameters have a major influence and should be listed.

√Thanks for your suggestions. We have added the necessary printing parameters: “During the LDED process, the laser power was 3 kW, the scanning speed was 900 mm/min, and the powder feed rate was 50 g/min. In order to reduce the anisotropy, the laser scanning path with a 45° angle was adopt-ed to build the parts layer by layer (Fig. 1a).”

  1. For other authors, they do not indicate the number of samples on which the experiments were performed. And if he gives the results in the text of Figure 4, then the results are given without statistical values.

√Thanks for your suggestions. The tensile properties along the horizontal direction have been tested and counted in previous studies [2]. This work focused on the fracture mechanism of the samples before and after annealing through in-situ tensile tests.

[2] Li F, Qi B, Zhang Y, et al. Effects of Heat Treatments on Microstructures and Mechanical Properties of Ti6Al4V Alloy Produced by Laser Solid Forming[J]. Metals, 2021, 11(2): 346.

  1. Figure 3a and 3d have very poor image quality.

√Thanks for your suggestions. We have improved the image quality of Figures 3a and 3d.

Figure 3. Various types of α phases including αGB, αw and α colony. (a-c) as-built sample, (d-f) annealed sample.

  1. There are minor typos in the text, such as: the name of the Joel IT 300 lv microscope is not in brackets like the rest of the instruments. Page 6 is at Fig. 7 is written in small letters.

√Thanks for your suggestions. We have carefully improved the quality of the text and language.
